# RNAi-Based Antiviral Innate Immunity in Plants

**DOI:** 10.3390/v14020432

**Published:** 2022-02-20

**Authors:** Liying Jin, Mengna Chen, Meiqin Xiang, Zhongxin Guo

**Affiliations:** Vector-Borne Virus Research Center, State Key Laboratory for Ecological Pest Control of Fujian and Taiwan Crops, College of Plant Protection, Fujian Agriculture and Forestry University, Fuzhou 350002, China; jinliying2021@163.com (L.J.); nm18838025837@163.com (M.C.); xiangmeiqin123@163.com (M.X.)

**Keywords:** virus, antiviral innate immunity, RNAi, small RNA, VSR

## Abstract

Multiple antiviral immunities were developed to defend against viral infection in hosts. RNA interference (RNAi)-based antiviral innate immunity is evolutionarily conserved in eukaryotes and plays a vital role against all types of viruses. During the arms race between the host and virus, many viruses evolve viral suppressors of RNA silencing (VSRs) to inhibit antiviral innate immunity. Here, we reviewed the mechanism at different stages in RNAi-based antiviral innate immunity in plants and the counteractions of various VSRs, mainly upon infection of RNA viruses in model plant Arabidopsis. Some critical challenges in the field were also proposed, and we think that further elucidating conserved antiviral innate immunity may convey a broad spectrum of antiviral strategies to prevent viral diseases in the future.

## 1. Introduction

The phenomenon of RNA silencing was first observed in plants in 1990 when the introduction of chalcone synthase transgene into petunia led to the suppression of endogenous homological genes [1,2]. In 1998, Andrew Fire and Craig Mello found that double-stranded RNAs (dsRNA) caused potent and specific interference in *Caenothabditis elegan* [3] and termed the phenomenon RNA interference (RNAi), which won the Nobel Prize in Physiology and Medicine in 2006 and opened up a revolution in the field of biology [3,4]. Numerous studies showed that RNAi was evolutionarily conserved in eukaryotes, and it regulated all aspects of biological events [5,6]. RNAi signaling was established with the identification of some key components in the pathway. It was found that noncoding small RNA is triggered by either self-complementary or double-stranded RNA (dsRNA) and functions as the true signal and specificity determinant of gene silencing. Primary duplex small RNA is, respectively, produced by processing hair-pin RNA or dsRNA to microRNA or siRNA using a specific Dicer [7]. Additionally, secondary small RNA usually needs to be produced through amplification by RNA-dependent RNA polymerase (RdRP) for efficient gene silencing. Duplex small RNA can then be methylated by HEN1 to increase stability and is loaded into effector Argonaute (AGO) proteins [8]. The passage strand of duplex small RNA is degraded by AGOs, and the guided strand will remain to form an RNA-induced silencing complex (RISC) [9]. RISC then targets complementary RNA by base-pairing to mediate degradation or translation inhibition in post-transcription gene silencing (PTGS) or induce transcription gene silencing (TGS).

There are two major classes of endogenous small RNA in plants: microRNAs (miRNAs) and siRNAs [10]. miRNAs and siRNA are, respectively, produced from the processing of hair-pin RNA or dsRNA by a specific Dicer. miRNA, usually 21 nt in length, is produced by Dicer-like 1 (DCL1) in Arabidopsis. Endogenous miRNAs usually mediate PTGS and play an essential role in all aspects of plant developmental processes. Furthermore, 21, 22, and 24 nt siRNAs are, respectively, produced by DCL4, DCL2, or DCL3 in Arabidopsis and also regulate various biological processes. While 21 and 22 nt endogenous siRNAs usually mediate PTGS, such as 21 nt tasiRNA involved in leaf morphogenesis, 24 nt endogenous siRNAs mainly mediate TGS through DNA methylation.

Upon pathogen infection, different sizes of pathogen RNA-derived siRNAs are also produced to induce RNAi-based antimicrobial immunity to confer host resistance [10,11]. RNAi-based antiviral defense was first discovered in plants [12,13]. It was then found to play a vital role in antiviral immunity in invertebrates [14] and mammals [15,16]. Based on the findings in transgene silencing and endogenous gene silencing, the function of DCLs, AGOs, and RDRs (RNA dependent RNA Polymerases) in antiviral immunity were characterized. Now it is recognized that RNAi-based antiviral innate immunity will be induced to prevent aggression of all kinds of RNA or DNA viruses in almost all eukaryotes (Figure 1). On the other hand, many viruses, especially pathogenic viruses, evolve to encode VSRs to attack different steps of the RNAi-based antiviral pathway (Figure 1). The prevalence of VSR contributes to viral epidemics; it also blinds our appreciation to RNAi-based antiviral innate immunity. In addition, VSR also hindered us using classical genetic screen to identify new regulators in the antiviral pathway for decades until recently an efficient genetic method was developed to bypass the barrier [17,18].

Here, we will review the perception of viral RNA and the initiation of RNAi-based antiviral defense, production, and amplification of viral siRNA (vsiRNA) and the functions of antiviral RNAi effector Argonautes, with an emphasis on the recent progress in the field, challenging the existing questions on model plant Arabidopsis. Some VSRs of plant viruses and their functions were also summarized for a better understanding of the arms race between plant hosts and viruses. We apologize that some research progresses in the field may not be included.

## 2. Perception of Viral RNA and Initiation of RNAi-Based Antiviral Defense

Viruses are almost the smallest organisms on Earth, with a classical structure in which genetic material, RNA or DNA, is packaged in coat protein. Viruses need to be paralyzed in the host and propagated using the materials and energy of the host. Unlike other microbial pathogens, pattern-recognition receptors (PRRs) are not found to perceive viruses on the cell membrane of the host. After virus enters the host cell, double-strand viral RNAs will be produced during viral replication through viral replicases, host Dicer proteins will perceive and dice the double-strand viral RNAs to produce 21–24 nt lengths of vsiRNA. Thus, Dicer protein may be regarded as a viral PRR used to initiate the RNAi-based antiviral pathway.

Dicer belongs to the RNaseIII-like family and has highly conserved endonuclease in eukaryotes [19]. In Arabidopsis, there are four Dicer-like proteins (DCL): DCL1, DCL2, DCL3, and DCL4. They all contain five domains which are DExD-helicase, helicase-C, domain of unknown function 283 (DUF283), Piwi/Argonaute/Zwille (PAZ) domain, two tandem RNase III domains, and one or two dsRNA-binding domains (dsRBDs) from the N-terminus to C-terminus [20] (Figure 2). DCL3 does not have a helicase-C domain. In general, the helicase domain utilizes ATP hydrolysis to facilitate the unwinding of dsRNA [19,21]. The DUF283 domain was recently described to facilitate RNA–RNA base pairing and RNA-binding [22,23]. The PAZ and RNase III domains are vital for dsRNA cleavage, the PAZ domain recognizes the terminus of the dsRNA and RNase III domains and cuts one of the strands of dsRNA, and the distance between the PAZ domain and RNase III domains is determined by the length of the products [22,24]. The dsRBD domain facilitates dsRNA binding and also serves as a nonclassical nuclear localization signal [23].

DCL1 is mainly involved in the biogenesis pathway of 21 nt micro-RNAs (miRNA), which play essential roles in all aspects of plant development and plant responses to environmental stimuli [25]. DCL1, DsRNA binding protein 1(DRB1) (also known as HYPONASTIC LEAVES 1, HYL1), and SERRATE (SE) form nuclear dicing bodies to recognize the hair-pin structure of pri-miRNA, and sequentially cut pri-miRNA to precursor miRNA (pre-miRNA) and pre-miRNA to mature miRNA [21,26]. Recently it was reported that phase separation of SE drives dicing body assembly and promotes miRNA processing by DCL1 in Arabidopsis [27]; SE-Associated Protein 1 also promotes miRNA biogenesis by modulating pri-miRNA splicing, processing, and stability [28]. Therefore, loss-of-function DCL1 mutants, embryonic lethal, and even hypomorphic mutant DCL1 also showed pleiotropic developmental defects due to disrupted miRNA biogenesis [29]. DCL1 may indirectly function in RNAi-based antiviral innate immunity through controlling the biogenesis of some miRNAs. It was reported that miR168 negatively regulates AGO1 accumulation in plants [30], and miR482 or miR6019/miR6020, respectively, decrease antiviral resistance of R-genesin tomato or tobacco [31,32]. DCL1 could also promote the other DCLs mediated biogenesis of vsiRNA [33,34].

DCL2 is responsible for processing exogenous double-stranded RNA (dsRNA) molecules or natural antisense siRNAs into 22-nt siRNAs in Arabidopsis [35,36]. However, DCL2 only subrogates to initiate RNAi-based antiviral innate immunity in Arabidopsis when the function of DCL4 is abrogated [37,38]. Interestingly, recent studies showed that massive endogenous 22 nt siRNAs can be accumulated when the cytoplasmic RNA decay pathway and function of DCL4 is defective; these siRNAs can trigger gene-specific and global translational repression and lead to pleiotropic growth disorders [39]. However, multiple orthologs of DCL2 exist in other plants and can evolve to possess functions. For example, a study demonstrated that DCL2b, one DCL2 homolog out of four in tomatoes, played a vital role against *tomato mosaic virus* (ToMV) infection by producing 22 nt vsiRNA in tomatoes [40].

DCL3 generates 24nt siRNAs to regulate RNA-dependent DNA methylation (RdDM) in transcriptional gene silencing (TGS) in Arabidopsis [41]. Recently, DCL3-pre-siRNA structure revealed that DCL3 used a positively charged pocket and an aromatic cap to, respectively, recognize the 5′-phosphorylated adenosine of the guide strand and the 3′ overhang of the complementary strand. The paired RNase III domains of DCL3 cut both strands of the RNA, determining the precise length of the product small RNA [42]. Endogenous 24 nt siRNAs are mainly produced from heterochromatin or the repeated-sequence rich region by DCL3 to maintain transposon silencing or genome stability; 24 nt siRNAs also repress transcription of the transgene or other exogenous DNA, such as DNA viruses [43,44,45]. It was reported that after the infection of the DNA virus, 24 nt vsiRNAs were produced to modulate DNA methylation and histone modification of viral DNA and to prevent viral infection [46,47].

DCL4 cleaves long endogenous dsRNA to produce 21 nt siRNAs, such as trans-acting siRNAs (ta-siRNAs), which are crucial for plant development [48,49,50,51]. Arabidopsis *dcl4* mutant showed a phenotype of elongated, downward curled rosette leaves [51,52] and augmented anthocyanin accumulation [53,54]. In RNAi-based antiviral innate immunity, DCL4 perceives and cuts long viral dsRNA to produce 21 nt vsiRNA to prevent viral infection, especially after the infection of RNA virus in Arabidopsis and other plants [49].

Although each DCL is responsible for the production of distinct small RNA, they may function redundantly or hierarchically in RNAi-based antiviral innate immunity. For example, DCL4 was regarded as an endogenous suppressor to repress DCL2-mediated production of 22 nt siRNAs [39,55]; however, DCL2 functions redundantly in RNAi-based antiviral innate immunity, especially when the function of DCL4 is compromised [56]. Thus, in the absence of both DCL2 and DCL4, virus titers would be dramatically increased [18,57,58]. Furthermore, 21 nt siRNA produced by DCL4 could also facilitate the RdDM pathway to defend against the infection of DNA viruses [41]. DCL2 and DCL3 need to function together in the defense against *potato spindle tuber viroid* [59]. In addition, DCL1 has the potential to produce 21 nt vsiRNA in the absence of DCL2, DCL3, and DCL4 [55,57].

DsRNA-binding (DRB) proteins are also required for the proper perception and dicing of viral RNAs by DCLs [60]. The Arabidopsis genome encodes five DRB proteins: DRB1/HYL1, DRB2, DRB3, DRB4, and DRB5 [61]. They contain one to three conserved dsRNA-binding motifs (dsRBMs), which consist of about 70 amino acids, forming *α-β-β-β-α* folds and two α-helices to interact with dsRNA [62,63]. DRBs interact with specific DCLs to execute their special function in small RNA biogenesis and antiviral defense [60]. For example, the interaction between DRB1 (HYL1)and DCL1 is required for miRNA biogenesis and is involved in selecting the guide strand loaded into RISC [64,65,66]. DRB4 interacts with DCL4 to form another kind of dicing body for the efficient processing of siRNAs. It was reported that the DRB4 mutation resulted in defective antiviral defense to the infection of *turnip yellow mosaic virus* (TYMV) [67]. The *drb3* mutant was hyper-susceptible to *cabbage leaf curl virus* (CaLCuV) and *beet curly top virus* (BCTV) infection, and the viral genome methylation was substantially reduced in *drb3* [47]. DRB2 was recently characterized as a wide-spectrum antiviral effector; overexpression of DRB2 decreased the accumulation of several different RNA viruses, including *tobacco rattle virus* (TRV), *tomato bushy stunt virus* (TBSV), *potato virus X* (PVX), and *grapevine fanleaf virus* (GFLV) [68].

## 3. Production and Amplification of vsiRNA

After the perception and dicing of viral dsRNA through DCL, primary vsiRNA will be produced. However, adequate secondary vsiRNA must be produced through amplification for efficient antiviral defense. Host RNA-dependent RNA Polymerase (RdRP) proteins are core factors for secondary vsiRNA amplification in plants and *Caenorhabditis elegan*. They exponentially generate viral dsRNA, which serves as DCLs substrates for vsiRNA biogenesis, probably using truncated viral RNAs as templates [18,69,70].

There are six RdRP proteins in Arabidopsis (RDR1 to RDR6). RDR1, RDR2, and RDR6 all share the C-terminal canonical catalytic DLDGD motif of eukaryotic RDRs and have orthologs in many plant species, while RDR3, RDR4, and RDR5 share an atypical DFDGD amino acid motif in the catalytic domain [71]. RDR1, RDR2, and RDR6 are well demonstrated to control RNAi-based antiviral innate immunity in Arabidopsis, although the function of the tandem-repeated RDR3, RDR4, and RDR5 in the Arabidopsis genome are not identified.

RDR1 can be induced by the infection of the virus [72], viroid [73], or salicylic acid treatment [74]. It was found to amplify 21 nt or 22 nt vsiRNA in RNAi-based antiviral innate immunity, especially upon the infection of RNA viruses. RDR1 does not regulate either endogenous siRNA biogenesis or plant development. However, it was found that RDR1 mediated the production of virus-activated endogenous siRNA (vasiRNA), a novel class of host siRNAs that may contribute to antiviral defense in plants [72].

RDR6 is constitutively-expressed in various tissues in Arabidopsis. RDR6 not only promotes RNAi-based antiviral innate immunity by mediating vsiRNA biogenesis, especially after the infection of RNA viruses, it also controls plant development by mediating the biogenesis of endogenous siRNA, such as tasiRNAs [75,76]. RDR6 usually forms siRNA bodies with a suppressor of gene silencing 3 (SGS3) to cooperatively function in the processes [77,78,79,80]. Thus, the *rdr6* mutant and *sgs3* mutant display the same defects in antiviral defense and development [78,81,82,83]. Interestingly, RDR6 and miR472 may also negatively regulate PAMP-triggered immunity (PTI) and effector-triggered immunity (ETI) through the post-transcriptional control of disease resistance genes [84] and contribute to double-strand break formation in meiosis in other plants [75]. Besides, rice (Oryza sativa) RDR6 plays an antiviral role in the defense against *rice stripe virus* (RSV) [85].

RDR2 mainly associates with Pol IV to form a complex for transcribing short dsRNA precursors, which are cleaved by DCL3 to produce 24 nt siRNAs for directing DNA methylation [86,87,88,89], although RDR2 might also be able to generate 23 to 27 nt small RNAs from MIR genes to mediate DNA methylation [90]. It was reported that RDR2, Pol IV, and DCL3, core components in the RdDM pathway, mediated 24 nt vsiRNA production and played major roles against the infection of DNA viruses, such as geminiviruses [44,45]. Interestingly, 21 nt vsiRNA amplified through RDR1 and RDR6 could also facilitate the RdDM pathway and contribute to plant defense against DNA viruses [91,92].

Several novel factors involved in the amplification of secondary vsiRNA were also recently discovered. Antiviral RNAi-defective 1(AVI1)/aminophospholipid transporting ATPase 2 (ALA2), ALA1, and AVI2 were identified through a robust forward genetic screen using a cucumber mosaic virus (CMV) mutant in which start codons of VSR-2b were mutated [17,93,94,95]. In the *ala1/ala2* or *avi2* mutant, production of secondary vsiRNAs was dramatically reduced. ALA1/ALA2 contain the typical P4-type ATPase structure (Figure 2) and may transport specific phospholipids across cellular membranes in plants. ALA1 and ALA2 could cooperate with RDR1 and RDR6 to promote secondary vsiRNA biogenesis, probably by defining the cellular localization of its substrate phospholipid [17,94]. AVI2 was also named as a New factor enhancer of *rdr6* 3 (ENOR3) since it was also identified through a genetic screen from *rdr6* background using another CMV mutant in which the 2b gene was deleted [96]. AVI2, a putative magnesium transporter in Arabidopsis, also promoted secondary vsiRNA biogenesis-dependent RDR1 and RDR6 [93]. Interestingly, calmodulin-binding transcription activator-3 (CAMTA3) was recently found to activate Bifunctional nuclease-2 (BN2) to stabilize AGO1/2 and DICER-LIKE1 and to activate RDR6 for the amplification of vsiRNAs [97].

RDRs and new factors such as ALA1/2 and AVI2 are widely conserved in plants and worms to ensure sufficient biogenesis of vsiRNA for efficient RNAi-based antiviral innate immunity. However, RDRs are absent in Drosophila, mice, and humans, in which a different mechanism was recently discovered for vsiRNA amplification through extrachromosomal circular DNA [21]. Whether the new mechanism also exists in plants or worms remains to be investigated.

## 4. Antiviral Function of RNAi Effector Argonautes

vsiRNA must be loaded to AGO effectors to form RISC, then targets complementary viral genomes to PTGS or TGS in RNAi-based antiviral innate immunity. Effector AGOs are evolutionarily conserved and widespread in eukaryotes, though absent in prokaryotes [98]. They were demonstrated to regulate a variety of biological progresses in plant development and the plant response to environmental stimuli [98,99,100,101,102,103], in addition to their functions in antiviral defense. Crystallographic studies showed canonical eukaryotic AGOs contain five domains called the N-terminal (N) domain, PIWI-ARGONAUTE-ZWILLE (PAZ) domain, middle (MID) domain, a PIWI domain, and a domain of unknown function 1785 (DUF1785) [104,105] (Figure 2). The N domain may block guide-target pairing beyond position 16, PAZ domain recognizes the 3′ end of sRNA, the MID domain anchors the 5′ phosphate of sRNA, PIWI domain possesses ribonuclease activity to slice target RNA [106,107,108], and the function of DUF1785 domains was recently shown to impair the perfect matched siRNA and miRNA duplexes [109]. Together, all domains facilitate the correct combination between sRNA and target RNA to ensure proper silencing.

Ten AGOs are encoded in Arabidopsis [110,111,112,113]. AGO1 and AGO2 are the main components of RNAi mediated antiviral immunity against RNA viruses [100]. AGO1 also functions as an effector of miRNA to regulate all aspects of plant development by modulating the expression of endogenous genes [114,115,116,117,118,119,120,121,122]. Thus *ago1* knockout mutants are lethal. Therefore, the function of AGO1 in RNAi-based antiviral innate immunity was only examined using hypomorphic AGO1 mutants, such as *ago1-27,* which still displayed severe developmental defects [123]. Unlike AGO1, AGO2 does not participate in regulating plant development, and the *ago2* mutant does not show defects in growth and development; AGO2 may solely regulate plant defense in Arabidopsis. It was reported that AGO2 prefers to bind vsiRNAs with 5′ terminal A and AGO1 prefers to U [124]. AGO2 is required for resistance to a broad spectrum of plant viruses [56,125,126,127,128]. It was also reported that the catalytical activity of AGO2 was necessary for local and systemic antiviral activity [125,127], while the *ago1* mutant with intact catalytical activity was susceptible to viral infection [123]. AGO2 is also involved in resistance against the phytopathogenic bacterium *Pseudomonas syringae* [129], and AGO2 binds with miR393b* and silences *MEMB12* to modulate exocytosis of antimicrobial PR proteins and increase antiviral activity [129]. Therefore, AGO1 and AGO2 may play distinct roles in antiviral defense in plants.

AGO4, AGO6, and AGO9 are the major effectors functioning in the RdDM pathway in Arabidopsis. AGO4, AGO6, and AGO9 were shown to bind 24 nt heterochromatic small interfering RNAs (het-siRNAs) and contribute to the RdDM pathway [130,131]. It was reported that AGO4 mainly combated the aggression of DNA viruses through modulating RdDM, as reported that *ago4* mutants was susceptible to the infection of BCTV due to the diminished hypermethylation on BCTV genome [47]. Surprisingly, *ago4* mutants are susceptible to several RNA viruses, such as *turnip crinkle virus* (TCV), *bamboo mosaic virus* (BaMV), and *plantago asiatica mosaic virus* (PlAMV) [132,133,134,135] through a mechanism independent of RdDM pathway [135].

As to other AGO effectors, AGO5 together with AGO2 participate in reducing *potato virus X* (PVX) systemic infection, while AGO5 only plays a secondary role when AGO2 is overcome in the initially infected leaves [136]. AGO7 (also known as ZIP) was found to be a crucial factor during TCV infection by the image-based disease analysis method [132]. AGO7 can also bind with miR390 and mediate the biogenesis of endogenous tasiRNA [137]. AGO10 cooperates with AGO1 and has a redundant role in protecting inflorescence tissues from the infection of *turnip mosaic virus* (TuMV) [125], besides its function in regulating shoot apical meristem development by bindingmiR165/166 [138].

Interestingly, more than 10 AGO orthologs were found in some important crops such as rice and tomato, with 19 orthologs in rice and 15 in tomato. They can evolve to have differential functions in antiviral defense and development. For example, when infected with *rice stripe tenuivirus* (RSV), RSV coat protein (CP) triggers JA accumulation and upregulates JA-responsive transcription factor JAMYB to directly bind to the AGO18 promoter to activate AGO18 transcription [139]. AGO18 will bind and sequester miR168, which increase the accumulation of AGO1 for the antiviral process [140]. On the other hand, AGO18 preferentially binds miR528 to upregulate the accumulation of ROS and resists virus infection [141]. Our unpublished data also shows that some AGO orthologs in tomatoes possess differential functions compared to Arabidopsis.

## 5. Viral Suppressors of RNAi

In the defense and counter-defense arm race between host plants and viruses, viruses evolve VSR proteins to inhibit RNAi-based antiviral innate immunity. VSRs target different steps of the RNAi-based antiviral pathway to counteract the conserved antiviral immunity (Table 1) [142,143].

A very common counteraction of VSRs is to impede vsiRNA amplification. For example, 2b of CMV, βC1 of *tomato yellow leaf curl China virus* (TYLCCNV), and P6 of *rice yellow stunt virus* (RYSV) interfere with the RDR1/6-dependent biogenesis of secondary vsiRNA [155,180,185]. The V2 of *tomato yellow leaf curl geminivirus* (TYLCV), P2 of RSV and P4 of *rice stripe mosaic virus* (RMSV) interacts with SGS3 to inhibit the biogenesis of secondary vsiRNA [181,182,189]. Geminivirus V2 protein was also found to disrupt the calmodulin–CAMTA3 interaction, which decreases the expression of RDR6 to reduce vsiRNA biogenesis [97].

Some VSRs were found to obstruct the perception or dicing of viral dsRNA. For example, CP of TCV could inhibit the dicing activity of DCL4 [55], and P6 of *cauliflower mosaic virus* (CaMV) interacts with DRB4 to block dsRNA binding [154]. Some VSRs, such as NSs of *tomato spotted wilt virus* (TSWV) and Hc-Pro of *potato virus Y* (PVY), also bind to long viral dsRNA, which could block the sensing or processing of the viral RNAs by DCLs. Some other VSRs could directly target vsiRNA to inhibit RNAi-based antiviral innate immunity. For example, P19, the well-known VSR of tombusviruses, binds and sequesters vsiRNA, while RNase III of *sweet potato chlorotic stunt crinivirus* (SPCSV) binds and mediate vsiRNA degradation, and HC-Pro of *zucchini yellow mosaic virus* (ZYMV) decreases vsiRNA stability by disturbing vsiRNA methylation by HEN1. Disturbing the antiviral function of effector AGOs is another strategy used by some VSRs. For example, P0 of *potato leafroll virus* (PLRV) can mediate AGO1 degradation, and 2b of CMV can interfere with AGO1 and AGO4 and disturb their functions.

Surprisingly, unlike the above VSRs that counteract RNAi-based antiviral innate immunity, other mechanisms were found to antagonize antiviral responses by some VSRs. Recently, a study showed that VSR p19 can interact with the receptor-like kinase (RLK) BARELY ANY MERISTEM 1 (BAM1) and BAM2 to inhibit the cell-to-cell movement of RNA silencing [193,194]. VSR C4 of *tomato leaf curl Guangdong virus* (ToLCGdV) can also interact with BAM1 to suppress PTGS and reverse methylation-mediated TGS [195]. In addition, accumulated evidence indicates autophagy-modulated plant–virus interactions [196,197]. It was reported that the cargo receptor NEIGHBOR OF BRCA1 (NBR1) could target HC-Pro to suppress the viral accumulation of TuMV [198]. However, γB, the VSR of *barley stripe mosaic virus* (BSMV), targeted AUTOPHAGY PROTEIN7 (ATG7) to disrupt the ATG7–ATG8 interaction and promote viral infection [199].

Now we know that almost all plant viruses, especially pathogenic plant viruses, possess one or more VSR. The existence of VSRs contributes to the successful aggression of viruses and viral epidemics; they also seriously hinder our appreciation of the indispensable antiviral innate immunity in plants and other eukaryotes.

## 6. Question and Perspective

RNAi-based antiviral innate immunity is recognized as a fundamental antiviral innate immunity in plants and animals; some key components functioning in the antiviral pathway were found and are well characterized. However, our understanding of the exact mechanism of the antiviral pathway is far from complete since only a few novel components in the pathway were found, and some critical questions or challenges in the field are waiting to be tacked. For example, (1) whether a specific cellular structure exists for vsiRNA biogenesis and amplification, and (2) whether/how vsiRNAs move over long-distances to confer systematic resistance. (3) Additionally, the impact of function differentiation on the translation inhibition or slicing of different AGOs and their functional localization in cells are largely unknown. (4) The true function and mechanism of RNAi-based antiviral innate immunity in important agricultural crops are elusive and (5) the application of RNAi-based antiviral innate immunity in agricultural practice needs to be further explored. Therefore, further studies are required to dissect the vital antiviral innate immunity, and progress in the field could eventually lead to finding novel strategies or methods to prevent specific and broad-spectrum viral diseases in plants and even humans.

## Figures and Tables

**Figure 1 viruses-14-00432-f001:**
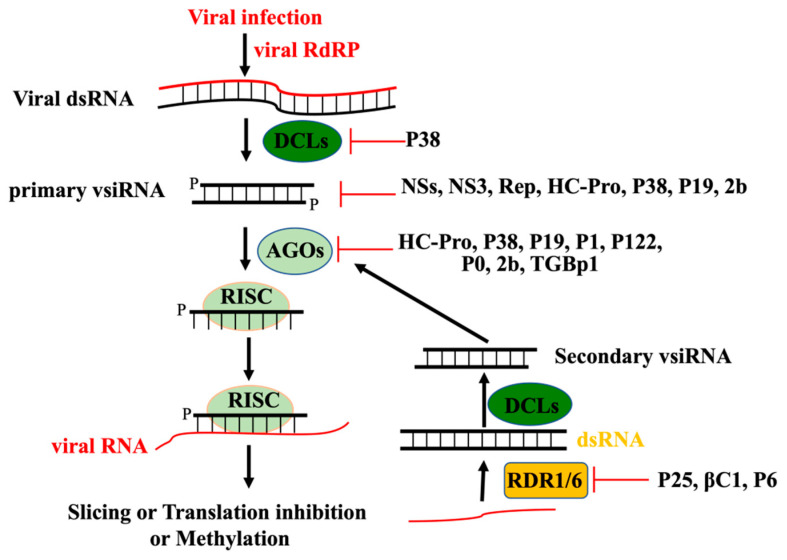
RNAi-based antiviral pathway in model plant Arabidopsis. After viral infection, the double-stranded viral RNA replicate intermediate will be perceived and processed into 21, 22, or 24 nt duplex primary viral siRNAs, respectively, by DCL4, DCL2, or DCL3. These 21 and 22nt primary viral siRNAs will then be uploaded into AGO1 or AGO2 to form RISC to mediate the slicing or translation inhibition of RNA viruses through PTGS. In contrast, 24nt vsiRNA will be uploaded to AGO4, AGO6, or AGO9 to form RISC to induce DNA methylation or histone modification through TGS to silence DNA viruses. Secondary viral siRNAs produced through amplification by RDR1/RDR6 or RDR2 are, respectively, required to enforce RNAi-based antiviral defense against RNA viruses or DNA viruses. Various VSRs of plant viruses target different steps to inhibit antiviral immunity.

**Figure 2 viruses-14-00432-f002:**
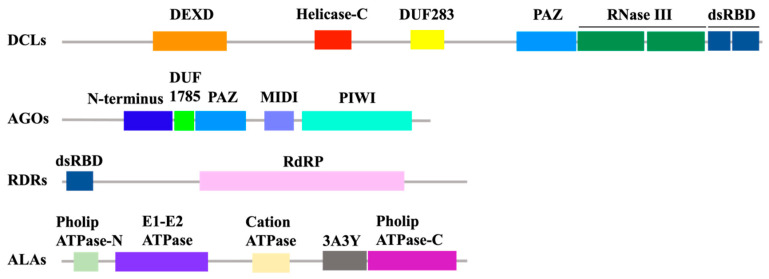
Protein structure diagram of DCLs, AGOs, RDRs, and ALAs. Function domains of each protein are shown with different colored boxes.

**Table 1 viruses-14-00432-t001:** Viral suppressors of RNA silencing (VSR) reported in plant viruses.

VSRs	Genus	Species	Viral Strategies to Suppress Antiviral RNA Silencing	Suppression of Systemic Silencing (YES/NO/ Unknown)
NSs	*Tospovirus*	TSWV	Binds with dsRNA (size-independent, long dsRNA, or dssRNA) [144,145]	YES
		GBNV	Affects miRNA biogenesis [146]	NO
NS3	*Tenuivirus*	RHBV	Binds dsRNA (size-selectively) [144]	Unknown
		RSV	Binds dsRNA (size-independent, recognizes minimum 9 bp or long dsRNA) [147] Interacts with DRB1 [148]	YES
Rep	*Mastrevirus*	WDV	Binds siRNA (size-selectively, 21 nt and 24 nt ds-siRNA and ss-siRNA) [149]	YES
P14	*Aureusvirus*	PoLV	Binds dsRNA (size-independent) [150,151]	Unknown
P15	*Pecluvirus*	PCV	Binds dsRNA (size-selectively) [151]	Unknown
γB	*Hordeivirus*	BSMV	Binds dsRNA (size-selectively) [151] The function of γb phosphorylation in regulating RNA silencing [152]	YES
P21	*Closterovirus*	BYV	Binds siRNA (size-selectively) [151] Blocks HEN1 methyltransferase [153]	Unknown
P6	*Caulimovirus*	CaMV	Interact with DRB4 [154]	NO
		RYSV	Blocks RDR6-mediated biosynthesis of secondary vsiRNAs [155]	YES
HC-Pro	*Potyvirus*	TEV	Binds dsRNA (size-selectively) [151] Inhibits AGO1 (by inducing miR168) [156] Interacts with RAV2 to block the activity of primary siRNAs [157]	Unknown
		TuMV	Interacts with HEN1 [158]	Unknown
		ZYMV	Interacts with the HEN1methyltransferase and inhibits its activity [159]	Unknown
		PVA	Interacts with AGO1 [160] Mediates HEN1 methyltransferase by inhibiting the SMA activity in the methionine cycle [160]	YES
		PVY	Binds 21 nt and 22 nt sRNA [161]	Unknown
		SCMV	Down-regulated the accumulation of RDR6 mRNA and 3′ secondary siRNA [162]	Unknown
CP(P38)	*Carmovirus*	TCV	Binds dsRNA (size-independent, long dsRNA or ds-sRNA) [151] Inhibits DCL4 activity [55] Inhibits AGO1 (by inducing miR168) [156]	Unknown
P19	*Tombusvirus*	CymRSV	Binds dsRNA (size-selectively, only 21 nt ds-sRNA) [151] Inhibits AGO1 (by inducing miR168) [30]	Unknown
		TBSV	Interacts with miRNA and hair-pin RNA-derived siRNA [163] Inhibits miRNA methylation [164]	Unknown
		CIRV	Binds dsRNA (size-selectively, 20-22 nt) [165]	Unknown
P1	*Ipomovirus*	CVYV	Down-regulates dsRNA formation and suppresses local RNA silencing through its duplicated form, P1b [166]	Unknown
		SPMMV	Interacts with AGO1 [167]	Unknown
P122	*Tobamovirus*	TMV	Binds dsRNA (size-selectively, siRNA and miRNA) and mediates 3′methylation of small RNA [168] Inhibits AGO1 (by inducing miR168) [156,168]	Unknown
P0	*Polerovirus*	PLRV	Targets AGO1 and promotes AGO1 degradation [169]	YES
		SCYLV	Suppression of local and systemic dsGFP-PTGS [170], while the detailed mechanism is unknown	YES
		CLRDV	Mediate the decay of ARGONAUTE proteins [171]	YES
		CYDV	Effects on secondary siRNA production and AGO1 stability [172]	YES
		BWYV	Acts as an F-box and targets AGO1 to modulate gene silencing [173,174]	NO
	*Enamovirus*	PEMV-1	Acts as an F-box and targets AGO1 for degradation [175]	YES
TGBp1 (P25)	*Potexvirus*	PVX	Interacts with AGO1 and promotes AGO1 degradation through the proteasome pathway [176]	YES
		PLAMV	Interacts with SGS3 and RDR6 [177]	Unknown
RNase III	*Crinivirus*	SPCSV	Cleaves 21-24nt vsiRNAs into 14 bp products and renders them inactive [178,179]	Unknown
βC1	*Begomovirus*	TYLCCNV	Repress RDR6 expression to inhibit secondary siRNA production [180]	Unkonwn
V2	*Begomovirus*	TYLCV	Interacts with SGS3 [181]	NO
V2	*Begomovirus*	CLCuMuV	Disrupts the calmodulin–CAMTA3 interaction [97]	Unknown
P4	*Rhabdovirus*	RSMV	Interacts with SGS3 [182]	Unknown
2b	*Cucumovirus*	CMV	Binds dsRNA (size-independent) [183] Interacts with AGO1 and AGO4 [184,185] Inhibits AGO1 (by inducing miR168) [156,185]	YES
		TAV	Binds dsRNA (size-dependent) [186] Down-regulates the accumulation of RDR6 mRNA [162]	Unknown
V3	*Begomovirus*	TYLCV	Functions in both PTGS and TGS [187]	Unknown
VPg	*Potyvirus*	PVA	Interacts with SGS3 and mediates SGS3 degradation through the ubiquitination and autophagy pathways [188]	Unknown
P2	*Tenuivirus*	RSV	Interacts with SGS3 [189]	Unknown
Pns10	*Phytoreoviru*	RDV	Interferes with the perception of silencing signals in recipient tissues [190] Binds dsRNA (with 2 nt 3′overhangs) [190,191] Causes downregulated expression of RDR6 [190]	YES
Pns11	*Phytoreovirus*	RDV	Dependent its nuclear localization signal [192]	No

NSS (nonstructural protein), VPg (Viral genome-linked protein), TGBp1 (the first ORF of triple-gene block proteins encoded protein), HC-Pro (helper component-proteinase), Rep (replication initiator protein), P14 (PoLV ORF5-encoded 14-kDa protein), P19 (19-kDa suppressor protein), P15 (PCV RNA-1-encoded 15-kDa protein), γB (BSMV RNAγ encodes γb protein), P21 (21-kDa product of BYV ORF 8), P6 (ORF6-encoded protein), CP (coat protein), P122 (122-kDa replicase subunit), P0 (the first ORF poleroviruses), RNase III (dsRNA-specific class 1 RNase III endoribonuclease), VPg (the viral protein genome-linked). SAMS (S–adenosyl-L–methionine synthase). BSMV (*barley stripe mosaic virus*), BWYV (*beet western yellows virus*), BYV (*beet yellows virus)*, CABYV (*cucurbit aphid-borne yellows virus*), CaMV (*cauliflower mosaic virus*), CIRV (*carnation Italian ringspot virus*), CLCuMuV (*cotton leaf curl Multan geminivirus*), CLRDV (*cotton leaf roll dwarf virus*), CMV (*cucumber mosaic virus*), CVYV (*cucumber vein yellowing virus*), CYDV (*cereal yellow dwarf virus*), CymRSV (*cymbidium ringspot virus*), GBNV (*groundnut bud necrosis virus*), PEMV-1 (*pea enation mosaic virus-1*), PCV (*peanut clump virus*), PlAMV (*plantago asiatica mosaic virus*), PLRV (*potato leafroll virus*), PoLV (*pothos latent virus*), PVA (*potato virus A*), PVX (*potato virus X*), PVY (*potato virus Y*), RDV (*rice dwarf virus*), RHBV (*rice hoja blanca virus*), RSMV (*rice stripe mosaic virus*), RSV (*rice stripe virus*), RYSV (*rice yellow stunt virus*), SCMV (*sugarcane mosaic virus*), SCYLV (*sugarcane yellow leaf virus*), SPCSV (*sweet potato chlorotic stunt crinivirus*), SPMMV (*sweet potato mild mottle virus*), TAV (*tomato aspermy virus*), TBSV (*tomato bushy stunt virus*), TCV (*turnip crinkle virus*), TEV (*tobacco etch virus*), TMV (*tobacco mosaic virus*), TSWV (*tomato spotted wilt virus*), TuMV (*turnip mosaic virus*), TYLCV (*tomato yellow leaf curl virus*), TYLCCNV (*tomato yellow leaf curl China virus*), WDV (*wheat dwarf virus*), ZYMV (*zucchini yellow mosaic virus*).

## Data Availability

Not applicable.

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
