# Peer review of "RNAi-Based Antiviral Innate Immunity in Plants"

_viruses, 2022, doi:10.3390/v14020432_

Round 1

Reviewer 1 Report

The review manuscript entailed “RNAi-based antiviral innate immunity in plant” by

Jin et al. summarized the development, progress and the most advances in respects of RNAi based antiviral innate immunity in plant. The manuscript is well written and the figures are well illustrated. However, there are still some gaps and important reference were not cited.

  1. The authors mostly focus on viruses that infect dicots, did not cover viruses that infect important monocot crops.
  2. The authors comprehensive summarized the functions of key components of antiviral RNAi from dicot, but rarely mention monocot crops, especially how these components were regulated in transcriptional level.

3.There are some important reference were not cited.

1). Gao F. A Non-structural Protein Encoded by Rice Dwarf Virus Targets to the Nucleus and Chloroplast and Inhibits Local RNA Silencing. Science in China, Life Science. 2020, 63, https://doi.org/10.1007/s11427-019-1648-3.

2).Ren B, Multiple functions of Rice dwarf phytoreovirus Pns10 in suppressing systemic RNA silencing. Journal of Virology. 2010, 84: 12914-12923.

3).Cao X. Identification of an RNA silencing suppressor from a plant double-stranded RNA virus. Journal of Virology. 2005, 79: 13018-13027. (Selected as article with significant interest in the Spotlight of issue).

4).Yang Z. et al. Jasmonate signaling enhances RNA silencing and antiviral defense in rice. Cell Host & Microbe, 28:1-1, 2020.http://doi.org/10.1016/j.chom.2020.05.001

5).Yang Z. etal. Dissection of RNAi-based antiviral immunity in plants. Current Opinion in Virology, 2018, 32: 88-99.

6). Zhang X. Contrasting effects of HC-Pro and 2b viral suppressors from Sugarcane mosaic virus and Tomato aspermy cucumovirus on the accumulation of siRNAs. Virology 2008, 374: 351-360

7). Jiang Let al. RNA-dependent RNA polymerase 6 of rice (Oryza sativa) plays role in host defense against negative-strand RNA virus, Rice stripe virus. Virus Research. 2012, 163(2): 512-519.

Author Response

Response to Reviewer 1 Comments

Jin et al. summarized the development, progress and the most advances in respects of RNAi based antiviral innate immunity in plant. The manuscript is well written and the figures are well illustrated. However, there are still some gaps and important reference were not cited.

Response 1: The topic has been reviewed from different views by Jin et al. and other scientists.  The recent references below have been added. Thanks.

Niu, D., Hamby, R., Sanchez, J. N., Cai, Q., Yan, Q., & Jin, H. (2021). RNAs - a new frontier in crop protection. Current opinion in biotechnology, 70, 204–212.

Huang, C. Y., Wang, H., Hu, P., Hamby, R., & Jin, H. (2019). Small RNAs - Big Players in Plant-Microbe Interactions. Cell host & microbe, 26(2), 173–182.

The authors mostly focus on viruses that infect dicots, did not cover viruses that infect important monocot crops.

The authors comprehensive summarized the functions of key components of antiviral RNAi from dicot, but rarely mention monocot crops, especially how these components were regulated in transcriptional level.

Response 2: Sorry for that the review mainly focuses on research progresses in model plant Arabidopsis as mentioned. Some research progresses in monocot crops have been included. Thanks.

3.There are some important reference were not cited.

1). Gao F. A Non-structural Protein Encoded by Rice Dwarf Virus Targets to the Nucleus and Chloroplast and Inhibits Local RNA Silencing. Science in China, Life Science. 2020, 63, https://doi.org/10.1007/s11427-019-1648-3.

2).Ren B, Multiple functions of Rice dwarf phytoreovirus Pns10 in suppressing systemic RNA silencing. Journal of Virology. 2010, 84: 12914-12923.

3).Cao X. Identification of an RNA silencing suppressor from a plant double-stranded RNA virus. Journal of Virology. 2005, 79: 13018-13027. (Selected as article with significant interest in the Spotlight of issue).

4).Yang Z. et al. Jasmonate signaling enhances RNA silencing and antiviral defense in rice. Cell Host & Microbe, 28:1-1, 2020.http://doi.org/10.1016/j.chom.2020.05.001

5).Yang Z. etal. Dissection of RNAi-based antiviral immunity in plants. Current Opinion in Virology, 2018, 32: 88-99.

6). Zhang X. Contrasting effects of HC-Pro and 2b viral suppressors from Sugarcane mosaic virus and Tomato aspermy cucumovirus on the accumulation of siRNAs. Virology 2008, 374: 351-360

7). Jiang Let al. RNA-dependent RNA polymerase 6 of rice (Oryza sativa) plays role in host defense against negative-strand RNA virus, Rice stripe virus. Virus Research. 2012, 163(2): 512-519.

Response 3: All these references have been added. Thanks.

Reviewer 2 Report

This review article nicely covers each of the different players of the RNAi-based antiviral immunity in plants and the counter-defense responses of plant viruses.

Since this review would be part of a special issue on the State-of-the-Art Plant virus research in China, the authors should take the time to introduce the latest advancements/discoveries in the field of RNAi, with an emphasis on the contributions of research labs in China including the tools/ approaches that contributed to the breakthrough.

 Some sections of the reviews require a bit more details.

Main comments:

  • The authors need to include paragraph in the introduction on the diverse small RNAs (miRNAs, vsiRNA, ta-siRNAs etc..) and how they are related to the immune response and/or vsiRNA production
  • Figure 1 would be more instructive if it describes each of the different components of the RNAi pathway (factors and small RNAs) with their distinctive and/or overlapping functions in plant development vs immune responses as well as in transcriptional vs post-transcriptional silencing.
  • Line 90: describe how exactly the miRNA biogenesis is involved in immunity.
  • Line 183. What is the update on the role of RDR6 in meristem protection against viruses? Does the interference of the RDR6 function by RYSV P6 (line 269-270) result in the invasion of the plant meristem by the virus?
  • Line 188: was the use of the CMV mutant that allowed the discovery of the ALA1 and ALA2?
  • Line 190: does the deletion or mutation of ALA1 and ALA2 then sufficient to abolish vsiRNA production?
  • Line 202-204: is there an evolutionary advantage for RDR to be absent in those species?
  • Section: challenge and perspective. What are the current challenges in the advancement in discovery? Is it technical limitation?

Minor comments:

Line 55-57. Rephrase. Unclear

Line 60-61: rephrase. unclear

Author Response

This review article nicely covers each of the different players of the RNAi-based antiviral immunity in plants and the counter-defense responses of plant viruses.

Since this review would be part of a special issue on the State-of-the-Art Plant virus research in China, the authors should take the time to introduce the latest advancements/discoveries in the field of RNAi, with an emphasis on the contributions of research labs in China including the tools/ approaches that contributed to the breakthrough.

Response 1: Thanks for all comments and suggestion. Some research progresses from Labs in China have been included. Sorry if not all are included in the review.

 Some sections of the reviews require a bit more details.

Main comments:

  • The authors need to include paragraph in the introduction on the diverse small RNAs (miRNAs, vsiRNA, ta-siRNAs etc..) and how they are related to the immune response and/or vsiRNA production

Response 2: A paragraph has been added to introduce diverse endogenous smalls and their functions in plant (line 42-52).

There are two major classes of endogenous small RNA in plants: microRNAs (miRNAs) and siRNAs. miRNAs and siRNA are respectively produced from the processing of hair-pin RNA or dsRNA t by specific Dicer. miRNA, usually 21nt length, is produced by DCL1 in Arabidopsis. Endogenous miRNAs usually mediate PTGS and play essential roles in all aspects of plant developmental processes. 21, 22, 24nt siRNAs are respectively produced by DCL4, DCL2 or DCL3 in Arabidopsis and also regulate various biological processes. 21, 22nt endogenous siRNAs usually mediate PTGS, such as 21nt tasiRNA involved in leaf morphogenesis, and 24nt endogenous siRNAs mainly mediate TGS through DNA methylation.

  • Figure 1 would be more instructive if it describes each of the different components of the RNAi pathway (factors and small RNAs) with their distinctive and/or overlapping functions in plant development vs immune responses as well as in transcriptional vs post-transcriptional silencing.

Response 3: Different components of antiviral RNAi (DCLs, AGOs and RDRs) and their distinct and/or overlapping functions in the pathway has been described. (line 994-1007)

Fig.1, RNAi-based antiviral pathway in plant plantsArabidopsis. After viral infection, the double-stranded viral RNA replicate intermediate will be perceived and processed into 21, 22 or -24 nt duplex primary viral siRNAs respectively by DCL4, DCL2 or DCL3s. This 21, 22nt primary viral siRNAs will then be mainly uploaded into AGO1 or AGO2s to form RISC to mediate the slicing or translation inhibition of RNA viruses through PTGS, while 24nt vsiRNA will be mainly uploaded to AGO4, AGO6 or AGO9 to form RISC to then mediate the slicing or translation inhibition of RNA viruses mainly through 21nt or 22 nt siRNAs mediated PTGS, or induce DNA methylation or histone modification through 24nt siRNAs mediated TGS to silence DNA viruses. Secondary viral siRNAs are produced through amplification by RDR1/RDR6 or RDR2host plant RDRs are respectively required for enforce RNAi-based antiviral defense against RNA viruses or DNA viruses, to re-enforce the RNAi-mediated antiviral innate immunity. RDR1 and RDR6 mediate production of secondary viral siRNAs against RNA viruses, while RDR2 Various VSRs of plant viruses target different steps to inhibit the antiviral immunity.

  • Line 90: describe how exactly the miRNA biogenesis is involved in immunity.

Response 4: DCL1 may indirectly regulate antiviral RNAi by controling the biogenesis of some miRNAs. The antiviral mechnisms of some miRNAs have been described in line 110-113.

It was reported that such as miR168 which negatively regulatesd AGO1 accumulaton in plants [31], and miR482 or miR6019/miR6020 respectively decreased antiviral resistance of R-genes,in tomato or tobacco [32].

  • Line 183. What is the update on the role of RDR6 in meristem protection against viruses? Does the interference of the RDR6 function by RYSV P6 (line 269-270) result in the invasion of the plant meristem by the virus?

Response 5: It was reported that RDR6 prevented meristem invasion of PVX-GFP or PSTVd in N. Benthamina (Schwach, et al., 2005; Di Serio, et al., 2010), but may not in Tomato (Naoi, et al., 2020). Whether inhibition of RDR6 function by RYSV P6 will lead to the meristem invasion of the virus need further investigation.

Schwach F, Vaistij FE, Jones L, Baulcombe DC. An RNA-dependent RNA polymerase prevents meristem invasion by potato virus X and is required for the activity but not the production of a systemic silencing signal. Plant Physiol. 2005; 138:1842–1852.

Di Serio F, Martínez de Alba AE, Navarro B, Gisel A, Flores R. RNA-dependent RNA polymerase 6 delays accumulation and precludes meristem invasion of a viroid that replicates in the nucleus. J. Virol. 2010; 84:2477–2489.

Naoi, T., Kitabayashi, S., Kasai, A., Sugawara, K., Adkar-Purushothama, C. R., Senda, M., Hataya, T., & Sano, T. (2020). Suppression of RNA-dependent RNA polymerase 6 in tomatoes allows potato spindle tuber viroid to invade basal part but not apical part including pluripotent stem cells of shoot apical meristem. PloS one, 15(7), e0236481.

  • Line 188: was the use of the CMV mutant that allowed the discovery of the ALA1 and ALA2?

Response 6: The CMV mutant without VSR 2b provided an efficient tool to uncover novel components in the pathway like ALA1 and ALA2.

  • Line 190: does the deletion or mutation of ALA1 and ALA2 then sufficient to abolish vsiRNA production?

Response 7: In ala1/ala2 and avi2 mutant, production of secondary vsiRNA was dramatically reduced. This was added in line 214-215.

  • Line 202-204: is there an evolutionary advantage for RDR to be absent in those species?

Response 8: That is an interesting question open for investigation. 

  • Section: challenge and perspective. What are the current challenges in the advancement in discovery? Is it technical limitation?

Response 9: “challenge” was changed to “question”. Some challenge questions in the field were listed in the last paragraph (line 339-345).

(1) whether specific cellular structure exists for vsiRNA biogenesis and amplification? (2) Whether/How vsiRNAs move over long-distance to confer systematic resistance? (3) And function differentiation on translation inhibition or slicing of different AGOs and their functional localization in cell are largely unknown. (4) The true function and mechanism of RNAi-based antiviral innate immunity in important agricultural crops are elusive. (5) Application of RNAi-based antiviral innate immunity in agriculture practice need to be explored.

Minor comments:

Line 55-57. Rephrase. Unclear

Line 60-61: rephrase. Unclear

Response 10: This paragraph was changed to:

Here we will review the perception of viral RNA and initiation of RNAi-based antiviral innate defense, production and amplification of viral siRNA (vsiRNA) and functions of antiviral RNAi effector AGOs, with emphasis on the recent progresses in the field and challenge questions existing in model plant Arabidopsis, some VSRs of plant viruses and their functions were also summarized for better understanding the arms race between plant host and viruses. We apologize if some research progresses in the field may not be included. (line63-73)